# Exploring the Impact of Online and Offline Channel Advantages on Brand Relationship Performance: The Mediating Role of Consumer Perceived Value

**DOI:** 10.3390/bs13010016

**Published:** 2022-12-24

**Authors:** Yunyun Zhao, Xiaoyu Zhao, Yanzhe Liu

**Affiliations:** 1School of Business Administration, Northeastern University, Shenyang 110169, China; 2Economics and Management School, Wuhan University, Wuhan 430072, China

**Keywords:** channel advantage, consumer perceived value, brand relationship performance, omnichannel retail, decision-making process

## Abstract

As omnichannel shopping behavior becomes increasingly popular among consumers, how to leverage the respective advantages and synergies of online and offline channels to retain customers for a long time is an urgent issue for retailers to solve. The purpose of this study is to explore the key advantages of online and offline channels influencing the omnichannel shopping experience in the decision-making process, and investigate their impact on consumer perceived value and brand relationship performance, as well as the interaction effect of online channel advantages and offline channel advantages. This study identifies the key advantages of online channels (search convenience, customer-generated information richness, and social connection) and offline channels (direct product experience, sales-staff assistance, and servicescape aesthetics) through a qualitative study and relevant literature review. Then, the proposed research framework was tested using the structural model equation in AMOS and hierarchical regression techniques in SPSS utilizing data from 347 shoppers. The results show that all variables except customer-generated information richness have positive impact on consumer perceived value. Other than search convenience and customer-generated information richness, consumer perceived value mediates the effect of other variables on brand relationship performance. Additionally, the interaction effect of online and offline channel advantages positively impacts consumer perceived value.

## 1. Introduction

Retailers can now deliver customers with products and services via a variety of channels, including online websites, smartphone apps, social media, and physical shops, thanks to advances in digital technology [1,2]. To meet diverse needs, consumers expect to alternately use different channels and touchpoints to complete the whole decision-making process [3], which promotes the evolution of the traditional single channel to the omnichannel retail model [4]. However, most omnichannel customers are found to be ineffectively served in practice, and there is a large disparity between consumer needs and omnichannel service capability [5]. A consistent shopping experience across channels no longer satisfies consumers’ demand for omnichannel shopping, and access to a variety of benefits and experiences across different channels is what they value most [6]. This means that for retailers, leveraging and integrating different channel advantages in channel integration should be paid more attention because it brings customers a comprehensive and optimal shopping experience.

Omnichannel shopping behaviors (cross-channel behaviors from the consumer perspective) occur when consumers evaluate the purchase costs and benefits of different channels at each stage of the decision-making process [7]. Gathering information, evaluating alternatives, and purchasing, are three primary stages of the decision-making process [8], which occupy the most time and energy of consumers. Channel advantages (as channel attributes) affect consumers’ channel choices at different decision stages [7]. Consumers use diverse combinations of complementary channels as part of the shopping experience, with the aim of leveraging the different channel advantages to get the optimal experience, such as efficiency, convenience, information availability, customer service and sensory experience [9]. Therefore, it is essential for omnichannel retailers to identify the advantages of online and offline channels as well as how they complement one another. However, many researches have concentrated on the relative advantages of digital channels [10,11], whereas the advantages of offline channels have been ignored. Little research has been conducted on what are the respective advantages of online and offline channels that influence the customer experience, and how they work together to maximize customer value.

Reviewing the relevant literature of cross-channel integration, many studies concentrated on channel-services configuration, integration consistency, and channel interaction [12,13], while little research has been conducted on the integration of different channel advantages. The online channel advantages (ONA hereafter) (such as convenience, information availability, and accessibility at any time) can effectively reduce consumers’ search costs, evaluation efforts, and time costs [14]. The offline channel advantages (OFA hereafter) bring consumers more experience benefits [3], such as product experience, social experience and the servicescape experience. The purpose of channel integration is to help consumers minimize transaction costs and maximize experience benefits during the decision process [14]. Although little literature was found to explore the association between online and offline channel advantages and consumer perceived value (CPV hereafter), some important channel attributes considered as channel advantages were found to have a positive impact on CPV. Search convenience in webrooming behavior [7], portability in the m-commerce context [15], and greater information availability in multichannel retail [16] are found to be positively associated with CPV. The helpfulness of salespeople and the gratification of touch in luxury consumption [7] and social interaction in the omnichannel shopping context [17] have also proven to have an important role in enhancing consumer value. Currently, there is a lack of systematic identification of the advantages of online and offline channels and establishing the relationship between channel advantages and CPV. Especially, little is known about the impact of combining different channel advantages on CPV in omnichannel service systems, and it remains unclear if online and offline channel advantages complement one another in terms of their impact on CPV.

CPV could lead to positive brand relationship outcomes [18]. Importantly, the core goal of omnichannel retailing is to build lasting and strong relationships between consumers and brands that yield positive brand relationship outcomes (also known as brand relationship performance) [3]. However, most omnichannel studies focus on consumers’ intention to purchase or repurchase [6], there is limited attention to improving brand relationship performance (hereafter BRP) [18], particularly on investigating the impact of channel advantages on BRP. Considering that repurchase intentions do not reflect the consumer-brand relationship from multiple dimensions, it makes sense to focus on BRP in omnichannel retail context.

Therefore, based on the above discussion and key relevant literature (see Table 1), this study aims to investigate the following research questions:

RQ1. What are the key advantages of online and offline channels influencing consumers’ omnichannel shopping experience in the decision-making process?

RQ2. Which of these predicts customer perceived value and brand relationship performance better? 

RQ3. Will online channel advantages and offline channel advantages have a positive synergistic effect on consumer perceived value?

We address the above research questions in the three steps that follow. First, a review of the relevant literature on omnichannel shopping behavior, consumer decision process, channel advantage, consumer perceived value, brand relationship performance, etc. Second, the qualitative study (25 in-depth interviews) identifies the key advantages of online and offline channels that influence customer omnichannel shopping experience at three stages of the decision-making process. Third, based on the consumer value framework, a conceptual model of channel advantages-CPV-BRP was constructed and empirically tested. Finally, it was further investigated whether ONA and OFA have a positive synergistic effect on CPV.

The findings have important theoretical significance. First, this study identifies the key online and offline channel advantages that influence the omnichannel customer experience. As important as channel integration quality is to the omnichannel experience, the complementary combination of online and offline advantages reflects the belief that customers are value maximizers in the decision-making process. Second, this study focuses on the effects of both online and offline channel advantages on CPV. It also investigates the relationship between omnichannel advantages-consumer value-brand relationship, rather than being limited to the interaction between consumers and channels, and finds that perceived value mediates between some channel advantages and BRP. Finally, the beneficial synergy between ONA and OFA may raise CPV, this reaffirms the necessity of combining the complementary advantages of online and offline channels to enhance the omnichannel customer experience. Regarding the practical implication, we screen out the advantages of online and offline channels and underline the beneficial interaction effect of ONA and OFA, which could assist omnichannel retailers optimize their omnichannel systems and give consumers a holistic omnichannel shopping experience.

The rest of this research paper is structured as follows: Section 2 presents background literature. Section 3 explains the identification of ONA and OFA using a qualitative study. Section 4 contains the model framework and hypotheses development. Section 5 shows the methodology, and Section 6 explains the results of the data analysis. Section 7 contains discussions and implications. Section 8 presents limitations and future research.

## 2. Theoretical Background

### 2.1. Consumer Decision Process and Cross-Channel Behaviors in Omnichannel Retail

The consumer decision process is not always linear, and consumers may switch back and forth between various phases of the purchase process. Consumers spend the majority of their time and energy on three key stages of the decision-making process: obtaining information, evaluating options, and purchasing [8]. To improve the opportunity for interaction with customers, omnichannel retailers provide a number of accessible channels and touch points, including physical shops, smartphone apps, and social media. By evaluating personal needs, channel characteristics, benefits received and costs paid, consumers use multiple online and physical channels to complete transactions in a single order [21]. At the same time, many complex cross-channel shopping behaviors have evolved due to the consumers’ different channel choices at each stage. Webrooming (researching information online and making a purchase offline) and showrooming (searching and experiencing products offline and later purchasing offline) are two common cross-channel shopping practices. With the growth of mobile devices and the ease of access to the Internet, mobile-showrooming is an increasing trend [22]. Consumers can search and experience products in-store while using their mobile phones to look up additional information and complete their transactions. Many “research shoppers” use both online and offline channels to gather information and evaluate products, and choose one of these channels to transact; this cross-channel shopping behavior is known as “mixed search and online purchasing” (or offline purchasing) [23]. Exploring the channel-related factors that influence the cross-channel shopping experience is necessary and may help retailers leverage channel features to effectively intercept consumers at specific stages of the decision process and maximize their shopping value.

### 2.2. Complementary Advantages across Channels in Omnichannel Retail

Online and offline channels have different competitive advantages and costs. Choudhury and Karahanna [10] proposed convenience, trust, and information accessibility, as advantages of digital channels over physical channels. Mobility and convenience are also considered to be the relative advantages of mobile banking [24]. Savings in search costs and travel costs are notable merits of the online channel [25]. Nevertheless, e-privacy issues, delivery time and the lack of sensory information are challenges for consumers making online purchases. A physical channel makes a significant contribution to the omnichannel shopping value. The physical inspection of products, social interaction, and instant gratification, are the main advantages of physical stores that attract many customers to visit [25]. Providing consumers with a hedonic and aesthetic experience is also one of the key advantages of physical stores [26]. However, high travel costs (including fuel consumption and travel time), product carrying, and shopping time constraints, are also major barriers to consumers getting to offline stores [25].

Based on the weighing of the advantages and costs of different channels, consumers use them complementarily and adopt different cross-channel shopping patterns. Some consumers use online channels as their primary shopping channel and physical stores as a supplemental channel. Other consumers, however, may use the opposite shopping pattern. The purpose of using these channels complementarily to complete the shopping journey is to minimize the transaction costs and maximize the experience benefits. Information complementarity between online and offline channels has been demonstrated [27]. Electronic channels offer advantages in mobility and search convenience that physical stores cannot match. Online reviews that shoppers value in the pre-sale phase are not available in the physical channel. However, physical stores provide shoppers with many face-to-face services, such as multi-sensory contact with products, personalized consultation, and the perception of the servicescape aesthetics. Research has shown that physical stores have been applied to complement online services [28]. Therefore, we propose that offering cross-channel advantages complementarity could optimize the decision-making process and enhance consumer value. Integrating complementary channel advantages helps enterprises form sustainable competitiveness and gain more loyal customers.

### 2.3. Consumer Perceived Value (CPV)

CPV as a source of the core competitiveness provides a foundation for understanding consumer behavior and decision process in retailing [29]. Woodruff proposed that the product attributes, attribute performance, and results generated by the use of products that achieve or hinder customer goals determine consumers’ perceived preference and evaluation, namely customer perceived value [30]. Based on this view, Sheth proposed that CPV involves five components: social, functional, emotional, conditional, and epistemic [31]. Sweeney and Soutar constructed a three-dimensional framework, including social, emotional, and functional value [32]. Subsequently, the classification of utilitarian and hedonic value is also presented in research [15]. These models view CPV as diverse benefits and advantages derived from the product and service [18]. Differently, Zeithaml defined consumer value as the consumer’s overall assessment of achieving the purchase task after a trade-off between perceived benefits and costs from a comparative perspective [33]. This definition sees consumers as value maximizers in purchase process [16]. Therefore, this study adapted the definition of Zeithaml. We argue that in the omnichannel retail context, CPV is influenced by diverse benefits (e.g., information, performance, convenience, sociability, pleasure) and costs (e.g., time, transportation costs, search and evaluation effort, risk uncertainty) [34]. Consumers weigh the relationship between potential benefits and costs at each stage of the decision process, and choose the channel that maximizes value to maximize total value [16].

### 2.4. Brand Relationship Performance (BRP)

The consumer–brand relationship reflects the extent of connection between the brand and the consumers [35], and is an important prerequisite for companies to maintain long-term success [36]. More and more omnichannel retailers are doing their best to take steps to establish a solid and long-term relationship with consumers in order to induce many positive customer-brand relationship outcomes, especially in an increasingly competitive environment [37]. BRP refers to the outcomes resulting from the consumer-brand relationship, such as brand commitment, customer satisfaction, word of mouth, brand trust, willingness to recommend and repurchase behavior [20]. The crucial relationship performance outcomes for retail brands in the retail market have been identified as brand satisfaction and brand loyalty [38]. In this study, BRP outcomes included brand satisfaction and brand loyalty. Brand satisfaction represents a cognitive evaluation of the customer based on all brand experiences related to the purchase [39]. Brand loyalty behavior includes two core components of attitudinal loyalty (such as customer commitment) and behavioral loyalty (such as word-of-mouth and repurchase), which is consistent with the definition of Jahn and Kunz [20].

## 3. Exploring ONA and OFA from the Consumer Perspective

The qualitative study was conducted to conceptualize and identify the key advantages of online and offline channels in the decision-making process based on consumer perceptions through literature analysis (see Table 2) and 25 in-depth interviews.

### 3.1. Qualitative Study

We conducted 25 in-depth interviews in China between July and August 2021 and qualitatively investigated the main advantages of online and offline channels. The subjects (22–41 years old) of the interview are required to have at least two omnichannel shopping experiences within the last 6 months. To ensure that participants provided the right insights about cross-channel shopping, we examined whether they had a high-involvement purchase, and participants who purchased high-involvement products valued both the online product information and the offline physical experience [40]. We used a validated 7-point scale [41] to assess subjects’ level of product involvement, and participants answered whether the product they purchased was unimportant/important, irrelevant/relevant, or doesn’t matter/matters to me. According to the findings, all subjects had a high level of product involvement (M = 4.23, SD = 1.13).

Each interviewee received a description of the three phases of the decision-making process as well as the definition of omnichannel purchase behavior prior to the interview. The protocol of the in-depth interview was dominated by open-ended questions, which were conducive to eliciting participants’ viewpoints. These questions were designed around the theme of the omnichannel shopping experience and decision-making process, mainly including the channel advantages of online over offline, the channel advantages of offline over online, and which channel attributes affect the omnichannel shopping experience and the decision-making process, etc. Appendix A shows the respondents’ demographic information. We recorded and took notes on the interviews. The length of each in-depth interview was 20–40 min. After the interviews, the recordings were converted into text files.

The Strauss and Corbin coding method was adopted for data analysis [42]. In open coding, the interview text was conceptualized and categorized sentence by sentence, forming the initial concepts and subcategories. In axial coding, the subcategories with similarity are summarized and divided into six main categories. According to the relevant research literature on the advantages of online and offline channels (see Table 2), three key advantages (such as search convenience, customer-generated information richness, and social connection) of the online channel and three key advantages (such as direct product experience, sales-staff assistance, and the servicescape aesthetics) of the offline channel were identified. These advantages cover the information gathering, evaluating alternatives, and purchasing phase, related to utilitarian and hedonic benefits to the customer.

**Table 2 behavsci-13-00016-t002:** Literature summary.

Channel Advantage	Supportive Views	References
Search convenience	Online channels have more obvious advantages in terms of search products and information in general than physical channels.	[43,44,45]
	Search convenience is the relative advantage of electronic channels.	
Customer- generated information richness	One of the key advantages of encouraging customers to make online purchases is the abundance of customer review feedback.	[17,46,47]
	Online customers provide rich product-related information, which includes text, images, and videos, and enables them to make informed decisions.	
	Consumers value adequate, relevant, and detailed product information and shopping experiences shared by others, and online channels meet this demand better than offline channels.	
Social connection	Online channels have the unique function of sharing product links and exchanging information with offline acquaintances at any time, which has a merit in enhancing the online shopping experience.	[48,49,50]
	The seamless connection between social tools and e-commerce is a new highlight of the digital channel, which better meets the social needs of consumers when shopping online.	
Direct product experience	Physical contact with the product is one of the important strengths of offline stores.	[51,52,53]
	Direct product experience in physical stores can shorten the sensory distance and information distance compared to indirect product experience in online channels.	
	Physical stores have more advantages in providing a product experience.	
Sales-staff assistance	Access to specialized services and the product knowledge of the sales staff are advantages of physical stores because they help buyers better evaluate products.	[7,54,55]
	The lack of direct interaction with sales staff has been a major drawback of online stores.	
	Face-to-face interaction with service providers in offline channels provides consumers with more tacit information and personalized service than in online channels.	
Servicescape aesthetics	There are more rich tangible cues (e.g., lighting, color, facilities, and decor) to present beauty to consumers in the offline environment than online.	[26,56,57]
	In many innovative physical stores, the aesthetic enjoyment of the store environment is its outstanding advantage, which attracts many consumers to enter the store to buy products.	

To further refine these advantages, we designed relevant questions and issued questionnaires, for example: Do you agree that search convenience is an advantage of online channels over offline channels? A 7-point Likert scale was used to evaluate each topic (1 = strongly disagree to 7 = strongly agree). We received 107 valid responses. The results showed that search convenience (M = 4.617, SD = 0.879), customer-generated information richness (M = 5.342, SD = 1.120), and social connection (M = 5.173, SD = 1.275) performed better in online channels than offline channels. Meanwhile, direct product experience (M = 5.789, SD = 1.294), sales-staff assistance (M = 5.211, SD = 1.112), and the servicescape aesthetics (M = 4.992, SD = 1.122) performed better in offline channels than online channels.

### 3.2. Online Channels Advantages (ONA) from the Consumer Perspective

Websites, mobile stores, applets, and social stores are common touchpoints of online channels in information gathering, evaluating alternatives, and purchasing stages. They provide consumers with important advantages and benefits at lower costs as follows: efficiency, convenience, information, cost-saving and sociability, and many others [14]. By combing and classifying the relevant literature on the advantages of online channels (see Table 2) and referring to the qualitative interviews, we summarize the key advantages of online channels based on the merits they exhibit at different stages of consumer decision-making as: search convenience, customer-generated information richness, and social connection. First, online channels have become more convenient under the influence of digital technology, especially mobile stores [58]. In the information gathering stage, consumers can easily search for and identity products through a variety of assistive search tools [44] at anytime and anywhere. Second, product evaluation information is critical to decision-making [59], which can help consumers efficiently complete searches and evaluations [60]. Third, in the evaluation stage, the social connection function of online channels satisfies consumers’ demand for information exchange with offline friends [50], which greatly optimizes consumers’ decision-making process [48]. Thus, we argue that the three advantages are not an exhaustive collection of online channels but are vital factors in optimizing consumer decision-making process.

Search convenience refers to “the speed and ease with which consumers identify and select products they wish to buy” [61] (p. 52). The online channel offers a variety of assistive search tools to improve the performance of search convenience, such as easy-to-use navigation, product categories suitable for multiple scenarios, personalized recommendations, and customer interaction systems that allow consumers to obtain the information they need with just a few clicks [44]. Compared to going to offline stores to select and learn about products, online channels help consumers save the effort and time of travel and speed up the shopping decision process. Search convenience enables consumers to quickly search for and identify products and information relevant to their goals in a limited amount of time, which is one of the important factors inducing webroom behavior [58].

According to Daft [62], customer-generated information richness, as a dimension of information quality, refers to the ability of evaluation information generated by customers with shopping experience to change understanding [46]. In the pre-purchase stage, searching for online reviews is the main task for most consumers [63]. Advances in technology make it easy for customers to post evaluation information on shopping platforms in multiple formats, such as plain text, text + image, text + video, etc. [64]. According to media richness theory [62], different media types have different levels of information richness [46]. The ability to convey information cues in descending order is plain text, images, and video [65]. The use of multimedia posting product evaluation content increases the perceived information richness of other consumers [64]. Rich customer-generated information (e.g., containing both text and images or videos) is adequate, relevant, and simple [66]. The richer the information, the more it reduces uncertainty and induces people to make a decision [64].

Social connection refers to the ability of online channels to provide shoppers with the feeling of being connected to others offline. The social connection function embedded in the shopping platform facilitates interpersonal communication and information exchange between consumers and their offline friends by sharing product links [48], regardless of time and location. Considering that consumers live offline and value friends’ advice when making decisions [55], retailers implement links between online platforms and social media (e.g., WeChat), with the goal that consumers who are browsing the products receive offline social support. In practice, online consumers can share a product link they are interested in with friends through this social component on the shopping platform, and they discuss, communicate, and evaluate the product with each other before making a decision [67]. Social connection satisfies consumers’ needs for information acquisition [50], especially in the evaluation stage. Compared to offline channels, the function provides consumers with a combination of online shopping and offline social interaction [49].

### 3.3. Offline Channels Advantages (OFA) from the Consumer Perspective

Offline channels play an important role in information gathering, evaluating alternatives, and purchasing phases, especially by providing opportunities for experience [3]. Under the challenge of digital technology, as a key channel for retail innovation, the functions and format of physical stores are also undergoing great changes [26]. Changing consumer demands are driving the transformation of the physical store from a place of transaction to a place with brand appeal in terms of enjoyment, design, personalization, sensory experience, and customer service [3]. Many innovative physical stores have improved customer experience in terms of customer service, store environment, multisensory experience, and human interaction [3]. Our review of the literature on the advantages of the offline channel (see Table 2) found that direct product experience, sale-staff assistance, and the servicescape aesthetics were the main advantages based on the consumer decision-making process, and this was validated by the results of the qualitative interviews. First, in the information collection and evaluation stage, direct product experience not only helps consumers to obtain a large amount of diagnostic information, but also enjoy sensory experience, especially with sensory products [52]. Second, sales-staff assistance provides consumers with much professional knowledge and tacit information, which makes it easier to obtain useful information and evaluate alternatives [7,68]. Third, the servicescape aesthetics affects consumers’ experience in the whole shopping process [3]. Moreover, the servicescape aesthetic can assist consumers in determining service quality [69]. Thus, we argue that the three advantages are the core elements that differentiate offline channels from online channels and optimize the consumer decision-making process.

Direct product experience refers to physical contact with a product through multi-sensory, as opposed to indirect product experience, which is the virtual presentation of a product through various external sources in the absence of actual sensory cues [41]. In general, sensory information is one of the primary distinctions between online and offline [52]. Direct product experiences, as opposed to indirect experiences, provide consumers with more diagnostic information about product attributes such as softness, weight, and color for consumer decision making [51]. The direct product experience is the outstanding advantage of the offline channel, attracting most consumers to visit offline stores and spend more money [70]. Physical interaction with products is an important experience in the cross-channel customer journey, especially when purchasing products with experiential attributes.

Sales-staff assistance is defined that consumers receive specialized advice, personalized attention, and service from sales-staff, which can reduce transaction uncertainty [7]. In the omnichannel retail context, the role of in-store staff has been redefined, with many omnichannel retailers positioning the salesperson as a purchasing decision assistant or a professional consultant [17]. Getting help from salespeople in the purchase process is a high-quality social interaction [54] that enhances consumers’ willingness to buy and builds a stronger relationship [17]. Additionally, sales-staff assistance, as an advantage of physical stores, becomes one of the key drivers of cross-channel shopping, especially webrooming behavior.

Servicescape aesthetics refers to the overall impression of a visually attractive shopping environment. Servicescape is closely related to aesthetic design [56]. Consumer visual aesthetics are shaped by the look and functional design of the store environment [1], which contains rich physical elements (e.g., lighting, color, facilities, decor, design style, and store layout) and provides experience possibilities [56]. Aesthetically pleasing design is becoming increasingly important. Especially in a hedonic context, consumers want to enjoy the visual aesthetics of the physical environment, and shopping is not the only reason they visit a store [71]. Therefore, the offline servicescape offers consumers a more aesthetic visual appeal.

## 4. Theoretical Framework and Hypotheses

The current study focuses on the advantages of online and offline channels that influence the holistic omnichannel experience in the decision-making process and explores the impact of these channel advantages on CPV and BRP in omnichannel retail context. Based on the consumer value framework, this study developed the association between channel attributes and CPV for the purpose of minimize transaction costs and maximize experience benefits, and constructed the conceptual model of channel advantages-CPV-BRP. The research model included omnichannel advantages (as the independent variables), CPV (as a mediating variable), and BRP (as a dependent variable), and nine hypotheses were developed (see Figure 1).

### 4.1. Online Channel Advantages (ONA) and Consumer Perceived Value (CPV)

The CPV from online channels is mainly driven by convenience value, which is related to the speed of obtaining services or products effectively and efficiently [71]. The website and mobile APP provide consumers with a large variety of products to choose form, and a clear navigation system helps them easily collect information about products and the brand, and identify the product they want to buy [72]. Search convenience, as an outstanding merit of the online channel, saves the cost and time of travel and brings utilitarian value to consumers [43]. Further, the search process is simple, it requires only a few clicks, and this enjoyable operating experience brings consumer hedonic value [43]. Therefore, we hypothesize that:

**Hypothesis** **1a.***Search convenience in online channels is positively related to consumer perceived value*.

Customer-generated information richness has an informative effect on consumer decision [73], and also provides consumers with potential value at multiple stages of the decision-making process [74]. Goh demonstrated that the richness of customer-generated evaluation information positively affects user purchase behaviors [75]. When consumers search for and evaluate information, they incur processing costs. Rich evaluation information enables consumers to quickly and clearly understand important information related to product attributes and usage experiences [47,76], reducing information processing effort and time, which in turn increases their perceived value [74]. Accordingly, we hypothesize that:

**Hypothesis** **1b.***Customer-generated information richness in online channels is positively related to consumer perceived value*.

Social interaction with offline people in online channels gives consumers a connection experience [77], and reshapes the omnichannel shopping experience. The social connection function brings consumers social support in many aspects [50], such as information exchange, useful advice, and a feeling of fun [50]. Consumers can obtain information from offline friends on online shopping platforms without leaving their homes, increasing their perceived value in information acquisition and convenience [77]. Consumers may feel happy to obtain friends’ advice and decision recognition in a convenient way, and further gain hedonic value [48]. Therefore, we hypothesize that:

**Hypothesis** **1c.**
*Social connection in online channels is positively related to consumer perceived value.*


### 4.2. Offline Channel Advantages (OFA) and Consumer Perceived Value (CPV)

Direct product experience is an unmediated interaction between a consumer and a product through all five senses [78]. Multi-sensory contact would lead to consumers’ value perception of utilitarian benefits (e.g., more product information acquisition) and hedonic benefits (e.g., sensory stimulation and enjoyment from sensory experience) [51]. Information gathering is an essential step in consumer decision process [79]. Direct product experience enables consumers to obtain a great deal of vivid and concrete information in a short time, which increases their information attainment value [78] and induces the perception of utilitarian value. Meanwhile, physical contact with products results in a hedonic experience [80]; specifically, touch makes shopping more enjoyable and enhances consumer perceived hedonic value [41]. Hence, we hypothesize that:

**Hypothesis** **2a.**
*Direct product experience in offline channels is positively related to consumer perceived value.*


Sales-staff assistance is particularly important during gathering information and evaluating alternatives phase [81]. Access to personalized advice and expert product knowledge provided by salesperson without much search and evaluation efforts enhances consumers’ perception of utilitarian value (informational value) [81], especially when they are faced with complex decisions and a lack of expertise [17]. Meanwhile, it is a pleasure for consumers to be assisted by sales-staff in resolving shopping queries, obtaining useful information, and eliminating purchase uncertainty [82], which brings them hedonic value [54]. Hence, we hypothesize that:

**Hypothesis** **2b.**
*Sales-staff assistance in offline channels is positively related to consumer perceived value.*


Whether the servicescape is beautiful is highly correlated with personal perceived value [83,84]. Servicescape aesthetics can help consumers evaluate service quality effectively and enhance their perception of utilitarian value [85]. Specifically, the servicescape is the physical “packaging” of a brand that sends quality clues to customers and helps them evaluate products and brands in a short time, which further improves shopping performance [83]. A visually attractive servicescape can immediately provide customers with good sensory experience, affect their arousal, pleasure, and excitement [85], and positively increase their hedonic value [69]. Hence, we hypothesize that:

**Hypothesis** **2c.**
*Servicescape aesthetics in offline channels is positively related to consumer perceived value.*


### 4.3. Consumer Perceived Value (CPV) and Brand Relationship Performance (BRP)

CPV is an important antecedent that affects the quality of the consumer–brand relationship [39]. Perceived value promotes consumer engagement, which in turn leads to better consumer–brand relationship outcomes as follows: satisfaction, customer commitment, and repurchase behavior [86,87]. In multichannel retailing, the convenience value [18] and emotional value [39] that consumers receive from online channels ultimately influence the outcome of brand relationships in terms of satisfaction and commitment. In physical stores, consumers gain both informational and hedonic value through multi-sensory interactions with the brand, which can improve the brand relationship quality, further increase brand loyalty, and lead to continued purchase intention [88]. Accordingly, we hypothesize that:

**Hypothesis** **3.**
*Consumer perceived value is positively related to brand relationship performance.*


### 4.4. Mediating Role of Consumer Perceived Value (CPV)

The analysis above has shown that better BRP is correlated with higher CPV. The study also suggests that CPV will mediate the relationship between service convenience, customer-generated information richness, social connection, direct product experience, sales-staff assistance, and servicescape aesthetics and BRP. This proposition is based on the argument of the CPV framework proposed by [18]. This argument states that the concept of CPV connects perceived benefits and relationship, and these benefits reflect consumers’ consideration of brand relationship [18]. Applied to this study, perceived value is composed of many benefits formed by their perception of channel advantages, which in turn induces positive brand relationship outcomes. In the online environment, search convenience saves consumers a lot of shopping travel costs [89]; customer-generated content provides helpful information such as purchase suggestions, product evaluations, or user experiences for consumer decision-making [90]; and the social connection function enables online consumers to get advice from offline friends at any time [77]. These utilitarian benefits of the online channel, obtained at a lower cost, lead to high consumer satisfaction and loyalty that reflect the performance outcomes of consumer–brand relationship [38]. In the offline environment, direct contact with the product provides a pleasant sensory experience and diagnostic product information [41]; salespeople provide the specialized information and personalized service that consumers need [54]; and a visually appealing servicescape can evoke consumers positive emotions [91]. These positive emotional and utilitarian benefits are values that consumers perceive and could induce desired performance outcomes as follows: word of mouth, recommendation, and consumer commitment [38]. Hence, based on the above discussion, we hypothesize that:

**Hypothesis** **4.**
*Consumer perceived value mediates the relationships between brand relationship performance and (a) search convenience, (b) customer-generated information richness, (c) social connection, (d) direct product experience, (e) sales-staff assistance, and (f) servicescape aesthetics.*


### 4.5. Interaction Effect of ONA and OFA on Consumer Perceived Value (CPV)

Cross-channel integration is the integration of firm information, transaction data, and service design-related elements on the one hand [12], and the integration of different channel advantages on the other [92]. The online and offline channels are both complementary and synergistic in shaping the consumer shopping experience [93]. When omnichannel retailers maximize the complementary advantages of the online and offline channels, this facilitates a positive synergistic effect between the two channels. For example, consumers’ perceived convenience of online channels may influence their experience in offline channels [94], and similarly, perceived friendly customer service and a pleasant servicescape experience in offline stores may improve consumers’ evaluation and attitude toward the online channel. From this perspective, the synergistic effect positively affects the holistic shopping experience throughout the consumer’s journey [93], and ultimately leads to high CPV.

However, if the complementary of online and offline channel advantages is low and consumers’ needs for diverse and personalized experiences are not met, the two channels do not produce positive synergies and may even mutually diminish the shopping experience in one channel [93]. For example, consumers who experience unpleasant customer service in physical stores may have less trust in online channels and need to search for more information to reduce perceived risk, which in turn negatively affects the consumer experience in the online channel. In these cases, consumers pay more for costs (search time and energy) and receive fewer benefits (unpleasing offline experience), which ultimately leads to low CPV.

Therefore, we propose that the interaction effect of ONA and OFA has an impact on CPV. When the omnichannel retailer adequately provides both online and offline channel advantages, the positive interaction effect will enhance CPV. Hence, based on the above discussion, we hypothesize that:

**Hypothesis** **5.**
*The positive interaction effect of ONA and OFA enhances consumer perceived value.*


## 5. Methodology

### 5.1. Data Collection and Sample

The online cross-section survey was conducted through the platform of wjx. We used a variety of approaches to access the participants, such as distributing survey links on social platforms such as WeChat and Weibo with the help of friends’ social networks, and commissioning professional platforms (WJX). We began the survey by providing participants with a brief introduction of omnichannel shopping behavior and omnichannel retail brands, and then identified respondents with omnichannel purchasing experience using screening questions. Those who qualified were asked to evaluate online and offline advantages, CPV, and BRP, according to their omnichannel experience. The respondents were over the age of 18 years. At the beginning of the questionnaire, the following screening question was posed: “Have you ever had an omnichannel shopping experience?” If the respondents answered “yes”, they were asked to provide the brand name and complete follow-up questions; if the answer was “no”, the survey ended. We collected 462 questionnaires in the survey, and 347 valid ones were available for further data analysis after excluding questionnaires with incomplete information and less than the average response time. All respondents received rmb 6 as a reward. The demographic analysis profile of responders is reported in Table 3. The sample group tends to be young consumers and more female. This is consistent with the characteristics of China’s consumer population. The data from 7th China Census 2020 shows that the total number of post-1980s, post-1990s and post-2000s is over 600 million (45.6%), who are familiar with Internet technology, prefer individual and diversified shopping experiences, and are the main force of China’s consumer market. Specifically, women have more purchasing power, with Suning Finance 2020 data showing that women contribute more than 70% of household spending.

### 5.2. Measurement Scale

All measurements were adapted from previous studies and have been proven to be reliable and valid. Some wording was modified to fit the omnichannel shopping context. The 7-point Likert scale (1 = strongly disagree, 7 = strongly agree) was used for all the constructs in this study. Search convenience was measured using three items adapted from [95]. Customer-generated information richness was measured with three items adapted from [66,96]. Social connection was measured with three items adapted from [97]. The three items measuring direct product experience were from [98]. The three items measuring sales-staff assistance were from [99]. The four items measuring servicescape aesthetics were adapted from [85]. CPV was measured with four items from [100]. The BRP was measured with five items based on [20,38]. All items were translated into Chinese. The constructs and measures are shown in Table 4.

## 6. Data Analysis and Results

The collected data for this investigation was analyzed in two steps. Confirmatory factor analysis was used to first evaluate the measurement model’s validity and reliability. Then, the direct hypotheses and the mediation analysis were examined by structural equation modeling (SEM) with AMOS 21, and the interaction effect of ONA and OFA on CPV was verified by multi-regression analysis with SPSS 25.

### 6.1. Measurement Model

The results in Table 4 show that the Cronbach’s alpha values for eight variables are higher than 0.7, supporting the measurement scale’s reliability [101]. All study constructs’ average variance extracted (AVE) values are higher than 0.5, and all composite reliability values are higher than 0.7, indicating the measurement scale’s convergent validity [101]. Table 5 further shows that the square root of the AVE is higher than the correlation between any two latent components, supporting the discriminant validity [102]. There is no multicollinearity between the study variables, as evidenced by the greatest correlation coefficient between the study structures, which was 0.736, which is below the threshold value of 0.9 [103].

### 6.2. Common Method Bias

To account for the consequences of common method bias, we used a variety of techniques. First, in the procedural aspect, the order of items was scrambled, and the participants’ anonymity was guaranteed. Second, in the statistical test, the Harman’s one-factor test was used to check for the occurrence of common method bias. We discovered that all components were driven into one factor, which accounted for 80.539% of the variance overall while only accounting for 36.11% of the variance in the first factor. Consequently, common technique bias is not a serious issue [104].

### 6.3. Hypothesis Testing

According to the SEM findings, the model fit was good (χ^2^/df = 2.756, GFI = 0.86, CFI = 0.934, NFI = 0.900, TLI = 0.922, RMSEA = 0.071) [101]. The path analysis results (Hypotheses 1–3) in Table 6 showed that while customer-generated information richness (β = 0.037, *p* > 0.05) did not significantly affect consumer perception of value, search convenience (β = 0.255, *p* < 0.05), social connection (β = 0.256, *p* < 0.001), direct product experience (β = 0.133, *p* < 0.05), sales-staff assistance (β = 0.189, *p* < 0.01), and servicescape aesthetics (β = 0.176, *p* < 0.01) did. Hence, results supported Hypothesis 1a, Hypothesis 1c, Hypotheses 2a–2c, but rejected Hypothesis 1b. A possible explanation is that consumers focus on information gathering, learning, and evaluating in the information search stage. Massive quantities of online reviews often mean that consumers do not find useful information, ultimately increasing their decision-making burden [105]. Although consumers incur no financial cost, they exert relatively high time cost and personnel efforts. Some researchers have pointed out that many customers consider time to be more valuable than money. Although the online channel satisfies consumers’ demand for information, they do not consider it worthwhile to achieve the purchase task based on the trade-off between gain and pay. The BRP was significantly impacted by CPV (β = 0.820, *p* < 0.001), and Hypothesis 3 was supported.

Next, we used the AMOS SEM’s bootstrapping process to do the mediation analysis. To test the significance of the mediation effects, 5000 bootstrapped re-samples were used, along with a bias-corrected confidence level of 95%. Whether or not the indirect effect fell within the 95% confidence interval of 0 determined whether the mediation effect was considered significant. If 0 is not included, the mediation effect is significant. The results in Table 7 revealed that CPV played a full mediating role in the impact of social connection, sales-staff assistance, and the servicescape aesthetic on BRP. Hence, Hypotheses 4c, 4d, 4e and Hypothesis 8f were supported, but Hypotheses 4a and 4b were rejected. A possible explanation is that, according to Herzberg’s two-factor theory [106], as advanced digital technologies bring tremendous innovation to online channels, the convenience and information richness of online channels are seen as “health” factors for consumers, and reducing these perceived benefits will severely impact their shopping experience. That is, search convenience and customer-generated information richness do not differ significantly across brands, and increasing them does not lead to positive brand relationship outcomes through CPV. This has important management implications for omnichannel retailers in terms of maintaining customer relationships.

To examine (Hypothesis 5) the interaction effect of ONA and OFA on CPV, we employed hierarchical regression analysis with SPSS. In Step 1, the model included control variables. In step 2, the model added independent variables (online channel advantage and offline channel advantage). In step 3, we entered the interaction effect [107] into the model. The results were presented in Table 8, where the interaction effect of ONA and OFA on CPV was significant (β = 0.064, *p* < 0.05).

This study carried out an additional analysis. According to the findings of Table 6, customer-generated information richness does not significantly increase CPV. Given the synergistic relationship between offline and online channel advantages, consumers’ perceived high level of offline channel advantages might enhance their positive experience of evaluation information richness and ultimately increase perceived value. Therefore, this study further investigated whether consumer perceived offline channel advantages and perceived online customer-generated information richness have a positive interactive effect on CPV. This study used model 1 in PROCESS marco to test whether the three offline channel advantages moderate the relationship between customer-generated information richness and CPV, specifically whether customer-generated information richness positively affects CPV at high levels of offline advantage. The results show that direct product experience (effect = 0.0448, *p* < 0.05, LLCI = 0.0030, ULCI = 0.0867) and sales-staff assistance (effect = 0.0723, *p* < 0.05, LLCI = 0.0172, ULCI = 0.1273) interacted positively with customer-generated information richness on CPV. While SSA (β = 0.0499, *p* > 0.05, LLCI= −0.0001, ULCI = 0.0998) did not.

## 7. Discussion and Implication

Based on the consumer decision process and the consumer value framework, this study attempted to propose a complementary framework of online and offline advantages and explore their impact on CPV and BRP. First, we conducted a qualitative study. This study identified key online and offline channel advantages influencing the omnichannel shopping experience from the consumer perspective. Second, we explored whether these channel advantages enhance CPV and ultimately result in positive BRP, as well as the mediating role of CPV in the relationship between channel advantages and BRP. Finally, this study revealed that the positive interaction effect of ONA and OFA could increase CPV, which proves that the complementary advantages between online and offline channels are significant in enhancing the holistic omnichannel experience and optimizing the firm’s omnichannel service systems.

In conformity with the relevant literature on online and offline channel advantages, our qualitative research results showed that search convenience, customer-generated information richness, and social connection are key advantages of online channels over offline. By contrast, direct product experience, sales-staff assistance, and the servicescape aesthetics of offline channels are key advantages over online. The results also demonstrate that, in addition to customer-generated information richness, other analyzed variables positively influence CPV. In practice, online channels provide great convenience for consumers to search, evaluate, and meet consumer demand for social interaction with offline friends. In offline stores, especially innovative ones, direct contact with products through hearing, vision, touch, smell, and taste not only gives consumers a happy experience but also helps them form cognition and judgment about products. Sales personalized assistance improves the quality of information communication and shopping pleasure. An aesthetically pleasing shopping environment reduces consumers’ experience costs and conveys a signal of service quality. These advantages reduce transaction costs, improve experience benefits, and ultimately positively affect CPV.

Consistent with prior studies, this study demonstrated that CPV is a significant component in producing favorable brand relationship results. CPV is critical factor in achieving positive brand relationship outcomes. Additionally, CPV was found to play a mediating role in the impact of social connection, direct product experience, sales-staff assistance, and servicescape aesthetics on BRP. The interaction impact of ONA and OFA also favorably influenced CPV, which is another significant discovery.

### 7.1. Theoretical Implications

The three areas covered by this study, in terms of theory, are as follows: First, in the three primary stages of information gathering, evaluating alternatives, and purchasing, we identified the key online and offline advantages influencing the omnichannel shopping experience. We have not found any studies comparing and incorporating the main advantages of online and offline shopping into the omnichannel field. Studies on channel integration have generally concentrated on channel interaction, integration consistency, and channel configuration. Thus, our findings supplement the literature on channel integration.

Moreover, this study reveals the mechanism of how channel advantages impact BRP through CPV. CPV is identified as the key factor influencing consumer decisions and leading to a strong brand relationship. We found that online advantages effectively reduce the purchase costs, and offline advantages provide more experience benefits. Based on the value framework, the combination of online and offline advantages enhances CPV by reducing more purchase costs (mainly from online) and increasing more experience benefits (mainly from offline). The results provide new insights into improving the perceived value of the omnichannel customers and optimizing consumer decision process. Additionally, given that omnichannel retailing regards the emotional connection with consumers as its foundation, omnichannel retailers seek to build a strong, lasting relationship with a consumer, not just one or two transactions. This study recognizes the importance of BRP for omichannel retailers and focuses on the correlation between channel advantages- CPV- BRP, which addressed the gap in the relationship between channel advantages and BRP and deepened understanding of consumer omnichannel behavior.

Finally, using a holistic perspective of the impact of online and offline channel advantages on the omnichannel shopping experience, this study discovers the positive interaction effect of ONA and OFA on CPV in omnichannel context. The finding provides novel insights into the holistic omnichannel experience perspective; that is, the synergistic effect of online and offline channel advantages could increase consumer value. Meanwhile, the complementary combination of online and physical advantages, on the other hand, was discovered in this study to be just as essential to the omnichannel experience as channel integration quality. This is a topic worthy of further study.

### 7.2. Managerial Implications

This study also has a number of implications for omnichannel retail practitioners. The research results could be used to design and optimize the omnichannel system to maximize consumer benefits, engage consumers, and foster a stronger bond between consumers and business.

In this study, based on the consumer decision process, the advantages of online and offline channels were screened. In addition, we found that the combination of online and offline advantages eventually positively impacts the BRP. The findings also illustrate the importance of the interaction effect of ONA and OFA. The research results may guide enterprises to optimize consumers’ costs and benefits by using multiple complementary touchpoints, facilitate consumer decision process, improve a holistic shopping experience, and finally induce more positive brand relationship results.

The results show that convenience and information interaction in online channels are always the key factors of consumers’ concern. Especially when the COVID-19 virus was found to be spreading, customers in this city were less likely to go out socially, which increased their reliance on online purchases. Omnichannel retailers should provide diversified services for consumers with the help of technology, such as webcast shopping, virtual fitting rooms, and 3D virtual displays, to optimize the service process and reduce barriers to purchase. In addition, direct product experience, sales-staff assistance, and servicescape aesthetics in physical stores significantly improve CPV. We suggest that retail brands should encourage consumers to physically contact with the product by giving them a chance to see, hear, smell, touch, or taste it. Service employees should act as consultants to provide professional knowledge [47]. More pleasing visual clues should be presented in the decoration and layout of physical stores. These measures would give consumers information value and shopping pleasure, and ultimately improve CPV and positive outcomes of the brand relationship.

Another management implication of the study is the duality of customer-generated information. Consumers need a lot of product knowledge, post-purchase experience, and other information to help them make wise decisions. However, disorderly reviews and mixed word of mouth on online channels increase the decision-making burden [105]. Online channels satisfy consumers’ needs for information, but no consumer feels that shopping is worthwhile because it requires a lot of cognitive effort. As a result, brands can display distinct information on product pages, such as videos, reviews, and sales volume. These signs can quickly convey valuable information to consumers.

The synergistic effect between online and offline channels deserves more attention from omnichannel managers, and both consumers and businesses stand to gain a lot from it. In the omnichannel retailing context, brand owners should not only pay attention to the consistency of channel integration but also focus on the combination of complementary advantages. Only in this way can consumer value be maximized and a stronger consumer relationship be built.

## 8. Limitations and Future Research

Future investigations may be able to address some of the study’s weaknesses. First, the collected data in the survey was self-reported by consumers based on their memories of omnichannel shopping experiences. Consumer recall bias may result in inaccurate data. Future research could seek to measure actual behaviors through on-site surveys to improve the validity of the model. Second, our mediating variable is CPV, which is an overall view of achieving the purchase task. In this study, we could not observe the impact of each advantage on specific value types, such as functional value and hedonic value; researchers could aim to solve this problem in the future. Finally, omnichannel retail is common in countries beyond China, the site of our research. This research should be replicated in other countries to enhance generalizability.

## Figures and Tables

**Figure 1 behavsci-13-00016-f001:**
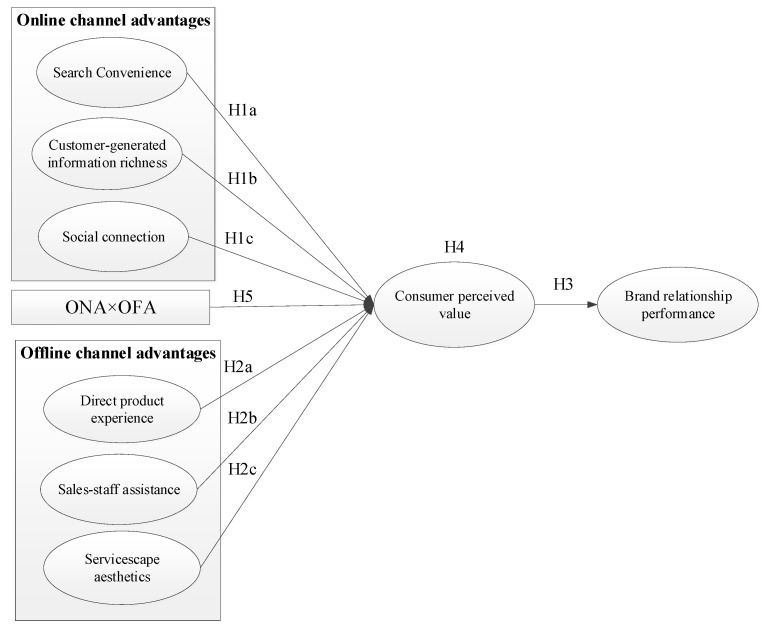
Research model. Note: ONA = online channel advantages, OFA = offline channel advantages.

**Table 1 behavsci-13-00016-t001:** Key relevant literature on channel advantages, CPV, and BRP.

Study	Channel	Focus	Channel Advantages		CPV	BRP
			ONA	OFA		
[12]	Omnichannel	The impact of omnichannel integrated quality on cross-buying behavior and customer value	No	No	Yes	No
[7]	Cross-channel	Exploring factors affecting luxury consumers’ webrooming behavior	No	No	Yes	No
[10]	Electronic channel	The effect of relative advantages of electronic channels on electronic channel adoption	Yes	No	No	No
[11]	Digital channel	How do relative convenience, relative advantage, perceived privacy, and perceived security of WeChat Pay influence continuous use intention	Yes	No	No	No
[19]	Multichannel	Evaluating the relative advantages between virtual worlds, websites and offline stores	Yes	Yes	No	No
[20]	Online channel	Impact of fan pages on customer-brand relationship	No	No	Yes	Yes
[18]	Online channel	Impact of customers’ perceived value on online channel satisfaction and loyalty	No	No	Yes	Yes
This study	Omnichannel	Impact of online and offline channel advantages on brand relationship performance	Yes	Yes	Yes	Yes

Note: ONA = online channel advantages, OFA= offline channel advantags, CPV = consumer perceived value, BRP = brand relationship performance.

**Table 3 behavsci-13-00016-t003:** Sample demographics.

Variables	Options	Percentage
Gender	Male	39.8%
	Female	60.2%
Age	18–24 years	40.9%
	25–34 years	46.6%
	35–44 years	11.1%
	More than 45 years	1.4%
Education	≤Junior College	5.6%
	Undergraduate	55.1%
	≥Postgraduate	39.3%
Shopping years in omnichannel	<1 years	11.8%
	1–2 years	28.1%
	2–4 years (including 2 years)	34.8%
	≥4 years	25.3%

**Table 4 behavsci-13-00016-t004:** Measures of key constructs.

Variables and Items	FL
Search convenience (α = 0.837, CR = 0.837, Mean = 5.787, AVE = 0.633)	
SEC1: Easy to understand and navigate in online channels.	0.874
SEC2: Find desired products quickly.	0.730
SEC3: Product classification is easy to follow.	0.776
Customer-generated information richness (α = 0.822, CR = 0.827, Mean = 5.534, AVE = 0.615.)	
In online channels, the customers with purchase experience:	
CGIR1: Provided sufficient evaluation information.	0.730
CGIR2: Provided easy to understand evaluation information in the form of text, images or videos.	0.830
CGIR3: Provide evaluation information at the right level of detail.	0.790
Social connection (α = 0.866, CR = 0.876, Mean = 5.329, AVE = 0.703)	
SOC1:I shared the link and views about the product with others offline through this link sharing tool.	0.716
SOC2: I benefited from others offline who received this shared link.	0.903
SOC3: I shared a common bond with others offline who received this shared link.	0.884
Direct product experience (α = 0.912, CR = 0.913, Mean = 5.634, AVE = 0.779)	
DPE1: I could see the product from all sides in the physical store.	0.899
DPE2: I could touch and feel the product in the physical store.	0.898
DPE3: I could physically inspect the product using multiple senses in the physical store.	0.849
Sales-staff assistance (α = 0.909, CR = 0.910, Mean = 5.465, AVE = 0.771)	
The sales-staff in physical store:	
SSA1: Gave me useful information of the product I wanted to buy.	0.844
SSA2: Provided friendly and personalized service to address my needs.	0.905
SSA3: Provided professional advice based on my needs.	0.884
Servicescape aesthetics (α = 0.914, CR = 0.914, Mean = 5.432, AVE = 0.727)	
SA1: The physical store is decorated in an attractive fashion.	0.857
SA2: The physical store displays its products in an attractive way.	0.859
SA3: The color schemes of the physical store are attractive.	0.839
SA4: In general, the physical store style is attractive.	0.855
Consumer perceived value (α = 0.925, CR = 0.912, Mean = 5.589, AVE = 0.755)	
CPV1: The time I spent shopping from this brand is worthwhile.	0.850
CPV2: The effort I spent shopping from this brand is worthwhile.	0.889
CPV3: The products and service of this brand I bought are valuable to me.	0.877
CPV4: The products and service of this brand meet my expectations and purposes.	0.858
Brand relationship performance (α = 0.891, CR = 0.892, Mean = 5.290, AVE = 0.625)	
BRP1: I am satisfied with the brand.	0.721
BRP2: I am an active supporter of this brand.	0.728
BRP3: I recommend this brand to other people.	0.860
BRP4: I intend to remain loyal to this brand in the future.	0.787
BRP5: I will continue purchasing this brand in the future.	0.847

**Table 5 behavsci-13-00016-t005:** Discriminant validity analysis.

Construct	1.	2.	3.	4.	5.	6.	7.	8.
1.SEC	0.796							
2.CGIR	0.732 **	0.784						
3.SOC	0.626 **	0.711 **	0.838					
4.DPE	0.663 **	0.655 **	0.586 **	0.883				
5.SSA	0.584 **	0.618 **	0.608 **	0.674 **	0.878			
6.SA	0.552 **	0.592 **	0.601 **	0.628 **	0.724 **	0.853		
7.CPV	0.714 **	0.728 **	0.733 **	0.717 **	0.733 **	0.706 **	0.869	
8.BRP	0.603 **	0.599 **	0.668 **	0.604 **	0.690 **	0.678 **	0.736 **	0.791

Note (s): ** *p* < 0.01; Boldfaced diagonal elements are the square roots of AVEs. SEC = search convenience, CGIR= customer-generated information richness, SOC = social connection, DPE = direct product experience, SSA = sales-staff assistance, SA = servicescape aesthetic, CPV = consumer perceived value, BRP = brand relationship performance.

**Table 6 behavsci-13-00016-t006:** Hypothesis testing.

Hypotheses	β	T Value	Results
H1a: Search convenience → CPV	0.255 *	2.412	Supported
H1b: Customer-generated information richness → CPV	0.037	0.285	Rejected
H1c: Social connection → CPV	0.256 ***	3.869	Supported
H2a: Direct product experience → CPV	0.133 *	2.204	Supported
H2b: Sales-staff assistance → CPV	0.189 **	2.930	Supported
H2c: Servicescape aesthetics → CPV	0.176 **	3.033	Supported
H3: CPV → BRP	0.820 ***	7.195	Supported

Note(s): *** *p* < 0.001 ** *p* < 0.01; * *p* < 0.05; CPV = consumer perceived value, BRP = brand relationship performance.

**Table 7 behavsci-13-00016-t007:** Mediation analysis.

Path		Coefficients	BootSE	Bootstrap 95% CIs		Medition
				Lower	Upper	
Service convenience → CPV → BRP	Direct	0.053	0.223	−0.354	0.415	No
	Indirect	0.209	0.166	−0.014	0.577	
Customer-generated information richness → CPV → BRP	Direct	−0.262	0.287	−0.756	0.199	No
	Indirect	0.030	0.208	−0.324	0.380	
Social connection → CPV → BRP	Direct	0.147	0.128	−0.082	0.377	Full
	Indirect	0.210	0.076	0.079	0.360	
Direct product experience → CPV → BRP	Direct	−0.092	0.089	−0.262	0.081	Full
	Indirect	0.109	0.068	0.020	0.242	
Sales-staff assistance → CPV → BRP	Direct	0.136	0.101	−0.057	0.339	Full
	Indirect	0.155	0.071	0.030	0.297	
Servicescape aesthetics → CPV → BRP	Direct	0.126	0.089	−0.041	0.298	Full
	Indirect	0.145	0.075	0.011	0.297	

Note (s): CPV = consumer perceived value, BRP = brand relationship performance.

**Table 8 behavsci-13-00016-t008:** Results of interaction effect of ONA and OFA on CPV.

CPV	Step 1		Step 2		Step 3	
	B	SE	B	SE	B	SE
Constant	5.824 ***	0.241	5.796 ***	0.123	5.786 ***	0.122
Sex	−0.037	0.114	−0.006	0.058	−0.010	0.057
Age	−0.264	0.239	0.014	0.122	−0.008	0.122
Education	0.537 *	0.223	0.163	0.125	0.171	0.124
Main effects						
ONA			0.488 ***	0.045	0.532 ***	0.048
OFA			0.463 ***	0.045	0.498 ***	0.047
Interaction effect						
ONA*OFA					0.064 *	0.027
R2	0.018		0.748		0.753	
ΔR2	0.018		0.731		0.004	
ΔF	0.108		0.000 ***		0.017 *	

Note (s): N = 347, * *p* < 0.05, *** *p* < 0.001.

## Data Availability

Data will be available on request.

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
