# Peer review of "Exploring the Impact of Online and Offline Channel Advantages on Brand Relationship Performance: The Mediating Role of Consumer Perceived Value"

_behavsci, 2022, doi:10.3390/bs13010016_

Round 1
Reviewer 1 Report
This article analyses a useful issue to increase firm’s sales and performance both through online and offline channel. For that purpose, authors developed a qualitative analysis conducting interviews (25) and they also use a structural model to determine the benefits of each channel.
In this article, the authors argue that their contribution consists on combining the analysis of the benefits of digital and offline channels to the consumer perceived value, given that each one also has costs. In addition, authors mention that cross-channel advantages are complementary and may optimize the decision-making process of customers.
In my opinion, this article exhibits a logical concern and it is correctly structured. It presents a good literature review and empirical research design is clear devoloped. However, I have a few recommendations to improve this article:
Given the number of consumers and firms with available online it would be important to indicate that the sample is a convenience sample and not probabilistic.
In addition, I suggest that some arguments should be developed regarding the costs of each channel online and offline, such as e-security or e-privacy for the former and fuel consumption or time wasted in the later.
Author Response
Dear reviewer:
Thank you for giving us the opportunity to submit a revised draft of the manuscript. We appreciate the time and effort that you dedicated to providing feedback on our manuscript and are grateful for the insightful comments and valuable improvements to our paper. We have studied the comments carefully and made corrections in accordance with the suggestions, which we hope meet your approval.
Point 1: Given the number of consumers and firms with available online it would be important to indicate that the sample is a convenience sample and not probabilistic.
Response 1: Thank you for this valuable suggestion. It was an oversight on my part not to mention convenient sampling. I have added the the procedure of convenience sampling on Page 14, Paragraph 4, Lines 1–4. The added text reads as follows: “The online cross-section survey was conducted through the platform of wjx. We used a variety of approaches to access the participants, such as distributing survey links on social platforms such as WeChat and Weibo with the help of friends' social networks, and commissioning professional platforms (WJX).”
Point 2: In addition, I suggest that some arguments should be developed regarding the costs of each channel online and offline, such as e-security or e-privacy for the former and fuel consumption or time wasted in the later.
Response 2: We agree that it is necessary to discuss both the advantages and costs of online and offline channels. We added the arguments for the costs of online channel on [Page. 5, Paragraphs 1, Line 6-7] read as follow: Nevertheless, e-privacy issues, delivery time and the lack of sensory information are challenges for consumers making online purchases. We added the arguments for the costs of offline channel on [Page. 5, Paragraphs 1, Line 12-14] read as follow: But high travel costs (Including fuel consumption and travel time), product carrying and shopping time constraints are also major barriers to consumers getting to offline stores [25].
Once again, thank you very much for your constructive comments and suggestions, which would help us in depth to improve the quality of the paper. Please let me know if you feel I have not accurately understood your suggestion. We are prepared to make additional revisions if you require them.
Reviewer 2 Report
Paper reviewed presents interesting topic and expresses advanced study on it. Theoretical gap is formulated and filled due research results. Structure of the paper is properly built. Authors formulate the state of problem in literature studies and gaps emerging from this analysis. They formulate research questions and discuss issues important for the paper content, making some indication about they utility for the paper. They formulate research model, formulate hypoteses, discuss variables. They explain research procedure by its stages, clearing the final outcome of each step. Research procedure and tools (statistical methods) are correctly used for the goals of the paper. They discuss widely results achieved theoretical and practical (managerial) context. They formulate interesting limitations and future direction of the research as well. Authors cite really wide group of scientific contemporary items of literature significant for the topic. It was real pleasure to read this paper and formulate my opinion.
Author Response
Dear reviewer:
Thank you for giving us the opportunity to submit a revised draft of the manuscript. We appreciate the time and effort that you dedicated to providing feedback on our manuscript and are grateful for your insightful comments on our paper. We also appreciate your acknowledgment.
Reviewer 3 Report
The paper deals with an interesting in the context of online and offline advantages. As a reviewer, I make several suggestions for improving the paper quality.
Introduction
The subject of the study and its objectives are presented properly. However, the paper can be improved for the work more synthesized. Furthermore, it should incorporate a description of the argument procedures used and the sequence of work. Contrary to what is stated in this section, this work does not present a true synthesis (or should be your intention) of the literature on the value generated the impact of online and offline channel advantages. Please revise this.
Theoretical Framework
All major relevant terms are discussed at work. The structure and paragraphs of this section can be improved, and the. Moreover, the research hypothesis will not be clearly exhibited linking with theoretical framework and literature review. Please revise this.
Methodology
The method selected for the empirical analysis is suitable for the study area. It is explained in a clear and understandable way. The research design is well organized. Well done.
Discussions and implication
The article discusses the existence of several existing conditions in the research model. It is appropriate to identify, explain and discuss its significance implications for the study. The discussions and implication clarify the accepted hypothesis and argument based acceptance.
- References
The number of references is acceptable and major works that are related to the issues cited. However, you are missing some important publications on the Behavioral Sciences in recent years.
English style
Overall, English style is suitable style.
Author Response
Dear reviewer:
Thank you for giving us the opportunity to submit a revised draft of the manuscript. We appreciate the time and effort that you dedicated to providing feedback on our manuscript and are grateful for the insightful comments and valuable improvements to our paper. We have studied the comments carefully and made corrections in accordance with the suggestions, which we hope meet your approval. The main corrections are in the manuscript and the responds to the reviewer’ comments are as follows (the replies are highlighted in red).
Introduction
Point 1: Furthermore, it should incorporate a description of the argument procedures used and the sequence of work.
Response 1: We appreciate receiving this valuable suggestion. I have added the description of the argument procedures on Page 3, Paragraph 2, Lines 1–6. The added text reads as follows: “First, a review of the relevant literature on omnichannel shopping behavior, consumer decision process, channel advantage, consumer perceived value, brand relationship performance, etc; Second, the qualitative study (25 in-depth interviews) identifies the key advantages of online and offline channels that influence customer omnichannel shopping experience at three stages of the decision-making process.”
Meanwhile, we added the the sequence of this research paper on Page 4, Paragraph 3. The added text reads as follows: “ The rest of this research paper is structured as follows: Section 2 presents background literature. Section 3 explains the identification of ONA and OFA using a qualitative study. Section 4 contains the model framework and hypotheses development. Section 5 shows the methodology, and Section 6 explains the results of the data analysis. Section 7 contains discussions and implications. Section 8 presents limitations and future research.”
Point 2: Contrary to what is stated in this section, this work does not present a true synthesis (or should be your intention) of the literature on the value generated the impact of online and offline channel advantages
Response 2: Thank you for your suggestion. Actually, I am sorry that I did not find studies that systematically identify online and offline advantages, and investigate their relationship to consumer perceived value. But I also made an effort to search the literature on channel attributes( and channel characteristics) positively associated with consumer value and made a brief summary. We added the literature review on [Page. 2, Paragraphs 3, Line 10-20] read as follow: “Although little literature was found to explore the association between online and offline channel advantages and consumer perceived value (CPV hereafter), some important channel attributes considered as channel advantages were found to have a positive impact on CPV. Search convenience in webrooming behavior [7], portability in the m-commerce context [15]and greater information availability in multichannel retail [16]are found to be positively associated with CPV. The helpfulness of salespeople and the gratification of touch in luxury consumption [7] and social interaction in the omnichannel shopping context [17] have also proven to have an important role in enhancing consumer value. Currently, there is a lack of systematic identification of the advantages of online and offline channels and establishing the relationship between channel advantages and CPV.”
Theoretical Framework
Point 3: The structure and paragraphs of this section can be improved, and the.
Response 3: We thank the reviewer for pointing this out. I have adjusted the structure of the Theoretical Framework and divided section 3 into section 3 (Exploring ONA and OFA from the consumer perspective) and section 4 (Theoretical framework and hypotheses).
Point 4: Moreover, the research hypothesis will not be clearly exhibited linking with theoretical framework and literature review.
Response 4: As suggested by the reviewer, we have added some arguments to the section 3 (theoretical framework) on [Page. 10, Paragraphs 4, Line 4-7], and the content reads as follows: Based on the consumer value framework, this study developed the association between channel attributes and CPV for the purpose of minimize transaction costs and maximize experience benefits, and constructed the conceptual model of channel advantages-CPV-BRP.
Meanwhile, we have added some arguments to the section 2 (literature review) on [Page. 5, Paragraphs 2] and [Page. 5, Paragraphs 3, Line 1-6], and the content reads as follows:
Online and offline channels have different competitive advantages and costs. Choudhury and Karahanna [10] proposed convenience, trust, and information accessibility as advantages of digital channels over physical channels. Mobility and convenience are also considered to be the relative advantages of mobile banking [24]. Savings in search costs and travel costs are notable merits of the online channel [25]. Nevertheless, e-privacy issues, delivery time and the lack of sensory information are challenges for consumers making online purchases. Physical channel make a significant contribution to the omnichannel shopping value. Physical inspection of products, social interaction and instant gratification are the main advantages of physical stores that attract many customers to visit[25]. Providing consumers with a hedonic and aesthetic experience is also one of the key advantages of physical stores[26]. But high travel costs (Including fuel consumption and travel time), product carrying and shopping time constraints are also major barriers to consumers getting to offline stores [25].
Based on the weighing of the advantages and costs of different channels, consumers use them complementarily and adopt different cross-channel shopping patterns. Some consumers use online channels as their primary shopping channel and physical stores as a supplemental channel. Other consumers, however, may use the opposite shopping pattern. The purpose of using these channels complementarily to complete the shopping journey is to minimize the transaction costs and maximize the experience benefits.
Point 5: However, you are missing some important publications on the Behavioral Sciences in recent years.
Response 5: We thank the reviewer for pointing this out. I have added 5 references from Behavioral Sciences on the paper.
Once again, thank you very much for your constructive comments and suggestions, which would greatly help us improve the quality of the paper. Please let me know if you feel I have not accurately understood your suggestion. We are willing to make additional revision if you require them.